EMBO
Molecular Medicine

# Interferon-beta signaling in retinal mononuclear phagocytes attenuates pathological neovascularization

Anika Lückoff[1], Albert Caramoy[1], Rebecca Scholz[1], Marco Prinz[2], Ulrich Kalinke[3] &
Thomas Langmann[1,*]

## Abstract

Age-related macular degeneration (AMD) is a leading cause of vision loss among the elderly. AMD pathogenesis involves chronic activation of the innate immune system including complement factors and microglia/macrophage reactivity in the retina. Here, we show that lack of interferon-β signaling in the retina accelerates mononuclear phagocyte reactivity and promotes choroidal neovascularization (CNV) in the laser model of neovascular AMD. Complete deletion of interferon-α/β receptor (Ifnar) using $Ifnar1^{-/-}$ mice significantly enhanced early microglia and macrophage activation in lesion areas. This triggered subsequent vascular leakage and CNV at later stages. Similar findings were obtained in laser-treated $Cx3cr1^{CreER}:Ifnar1^{fl/fl}$ animals that allowed the tamoxifen-induced conditional depletion of Ifnar in resident mononuclear phagocytes only. Conversely, systemic IFN-β therapy of laser-treated wild-type animals effectively attenuated microgliosis and macrophage responses in the early stage of disease and significantly reduced CNV size in the late phase. Our results reveal a protective role of Ifnar signaling in retinal immune homeostasis and highlight a potential use for IFN-β therapy in the eye to limit chronic inflammation and pathological angiogenesis in AMD.

**Keywords** age-related macular degeneration; choroidal neovascularization; interferon-beta signaling; macrophages; microglia

**Subject Categories** Immunology; Neuroscience; Vascular Biology & Angiogenesis

## Introduction

Age-related macular degeneration (AMD) is the leading cause of visual impairment among the elderly, often resulting in blindness (Augood *et al*, 2006). It occurs in two main clinical forms, the dry form with geographic atrophy and the wet or exudative form. The dry form is characterized by accumulation of cellular debris in the subretinal space, while the wet form typically presents with neovascular processes (Noel *et al*, 2007). Anti-VEGF medication with monoclonal antibodies or aptamers is presently the gold standard in the treatment of choroidal neovascularization (CNV) (Rofagha *et al*, 2013). However, inhibition of angiogenesis alone does not affect the cellular immunological events underlying CNV and also fails to be effective in the dry form of AMD (Jager *et al*, 2008). Furthermore, VEGF inhibition may cause a number of severe retinal and systemic adverse events (Sene *et al*, 2015). Therefore, there is an urgent need for identifying alternative approaches to treat AMD that is based on the underlying immunological pathogenesis.

Microglial cells, the resident immune cells of the CNS, play a key role in the initiation and perpetuation of chronic inflammatory events in the aging retina (Sierra *et al*, 2007; Damani *et al*, 2011). Furthermore, mononuclear phagocyte reactivity associated with subretinal migration is a common hallmark in human AMD and related mouse models (Gupta *et al*, 2003; Combadière *et al*, 2007; Levy *et al*, 2015). Consequently, signaling pathways and molecular targets that modulate microglia and macrophage activity represent attractive therapeutic tools for the treatment of degenerative and inflammatory diseases of the retina, including AMD (Langmann, 2007).

Type 1 interferon signaling through the interferon-alpha/beta receptor (Ifnar) is critically important for innate immune defense (Sadler & Williams, 2008). There is also strong evidence that interferon-β (IFN-β) has potent immunomodulatory functions on microglia and thereby limits autoimmunity in the CNS (Axtell & Steinman, 2008; Prinz *et al*, 2008). Likewise, mice lacking Ifnar or the IFN-β gene display extensive microglia activation and develop a more severe and chronic course of experimental autoimmune encephalomyelitis (EAE) (Teige *et al*, 2003; Prinz *et al*, 2008). Furthermore, IFN-β can block the production of neurotoxic superoxide radicals in microglia *in vitro* (Jin *et al*, 2007), and induction of

1 Retinal Immunology Laboratory (RIL), Department of Ophthalmology, University of Cologne, Cologne, Germany
2 Institute of Neuropathology and BIOSS Center for Biological Signaling Studies, University of Freiburg, Freiburg, Germany
3 Institute for Experimental Infection Research, TWINCORE Centre for Experimental and Clinical Infection Research, a joint venture between the Helmholtz Centre for Infection Research and the Hannover Medical School, Hannover, Germany
*Corresponding author. Tel: +49 221 478 7324; E-mail: thomas.langmann@uk-koeln.de

endogenous IFN-β by TLR3 or MDA-5 and RIG-I (Dann et al, 2011) ligands protects from EAE via an immunoregulatory pathway (Touil et al, 2006).

Here, we have studied the role of IFN-β and its receptor Ifnar in the laser-induced choroidal neovascularization model mimicking age-related macular degeneration in mice (Lambert et al, 2013). Previous studies have shown beneficial effects of IFN-β treatment on laser-induced CNV in rabbits (Kimoto et al, 2002) and monkeys (Tobe et al, 1995) by influencing the function of RPE and endothelial cells. Here, we demonstrate a pivotal effect of Ifnar/IFN-β signaling in retinal microglia and macrophages cells that reduce the inflammatory and angiogenic events and thereby limit the development of CNV lesions.

# Results

## Loss of Ifnar1 signaling promotes retinal microglia/macrophage reactivity and angiogenesis

We first determined whether Ifnar1 signaling affects the three key events in laser-induced CNV, (i) microglia/macrophage activation, (ii) vascular leakage, and (iii) neovessel formation. These three processes therefore were characterized in a kinetics analysis 3, 7, and 14 days after laser treatment of wild-type animals and $Ifnar1^{-/-}$ mice, respectively (Fig 1A). The confocal analysis of retinal flat mounts revealed that Iba1$^+$ cells with mixed ramified and amoeboid phenotypes were present at the lesion site 3 days post-laser damage (Fig 1B). In contrast, retinas from $Ifnar1^{-/-}$ mice predominantly revealed amoeboid-shaped cells in the laser lesion, indicating a more reactive cell population (Fig 1C). The quantification of amoeboid Iba1$^+$ cells from z-stack images then revealed significantly higher cell numbers in the laser spots of $Ifnar1^{-/-}$ mice ($14 \pm 1.9$ cells/spot) compared to control animals ($4.5 \pm 1.2$ cells/spot) (Fig 1D). The total number of Iba1$^+$ cells within the retina was not statistically different in both groups of animals (Appendix Fig S1A). To further define the cellular phenotype at the lesion sites, changes in their ramification and the length of processes were determined by counting the number of crossing points per individual cell using a grid image system (Chen et al, 2012). These observations also revealed a significant reduction in grid cross points in $Ifnar1^{-/-}$ retinas compared to controls, which clearly indicates a general shift in the morphology of microglia and macrophages (Fig 1E).

Given the strong influence of Ifnar1 deficiency on early mononuclear phagocyte activation, we next analyzed vascular leakage and CNV formation using fundus fluorescein angiography and lectin staining of RPE/choroidal flat mounts at different time points after laser lesion. We found that Ifnar1 deficiency had no major impact on CNV at day 3 and day 7 (Fig 1F, G, I and J), but resulted in a significantly higher vascular leakage at day 14 compared to controls (Fig 1H and K). Furthermore, lectin staining revealed a significant increase in the total CNV area in $Ifnar1^{-/-}$ mice compared to wild-type animals at 14 days after laser damage (Fig 1M–Q). We also analyzed retinal sections at day 3 and noticed prominent co-labeling of Iba1 and lectin in the subretinal area of $Ifnar1^{-/-}$ mice compared to wild-type controls (Fig EV1A and B). Moreover, in $Ifnar1^{-/-}$ mice reactive Iba1$^+$ cells persisted

longer in this area as choroidal flat mounts revealed considerable Iba1-lectin co-staining at day 7 post-laser treatment (Fig EV1C and D). These findings indicate that loss of Ifnar1 signaling in the retina triggers early microglia/macrophage reactivity that persists for several days and thereby negatively influences late outcome of CNV.

## IFN-β therapy attenuates mononuclear phagocyte reactivity and limits choroidal neovascularization

Having demonstrated that laser-induced CNV was enhanced in animals lacking Ifnar1, we next investigated the potential therapeutic effect of IFN-β. After laser coagulation, wild-type mice were systemically treated with 10,000 units recombinant human IFN-β every second day and CNV was analyzed after 3, 7, and 14 days, respectively (Fig 2A). IFN-β-treated animals displayed a significantly lower number of amoeboid Iba1$^+$ cells in the laser spots at day 3 when compared to untreated controls (Fig 2B–D). In addition, IFN-β therapy strongly influenced the phenotype of microglia/macrophages toward a highly ramified state characterized by more cellular processes (Fig 2E) without influencing the total number of Iba1$^+$ cells within the retina (Appendix Fig S1B). Furthermore, IFN-β treatment considerably improved disease outcome by reducing vascular leakage (Fig 2F–L) and CNV lesion size (Fig 2M–Q) 14 days post-laser damage. The analysis of retinal sections at day 3 (Fig EV2A and B) and of RPE/choroidal flat mounts at day 7 (Fig EV2C and D) showed much weaker Iba1-lectin co-staining in IFN-β-treated animals, indicating less pronounced immune reactivity associated with smaller CNV.

## Loss of Ifnar1 on resident microglia/macrophages affects their immunomodulatory potential and supports choroidal neovascularization

To establish whether Ifnar1 signaling in retinal microglia and potentially also long-lived choroidal macrophages directly modulates CNV formation, we used a specific targeting strategy by crossing $Ifnar1^{fl/fl}$ mice with $Cx3cr1^{CreER}$ mice (Goldmann et al, 2013). This mouse line carries a tamoxifen-inducible Cre recombinase under the control of the $Cx3cr1$ (fractalkine receptor) promoter (Yona et al, 2013). To be targeted by this genetic system, cells must express CX3CR1 at the time of tamoxifen application and be self-maintaining over a longer time period. Thus, tamoxifen-induced Cre recombinase expression is sustained in long-lived microglia, whereas short-lived myeloid cells lose Cre activity (Bruttger et al, 2015). We first tested microglia-specific targeting of $Cx3cr1^{CreER}$ mice that were crossed with $R26^{tomato}$ Cre reporter mice and treated them with tamoxifen. The confocal analysis of retinal flat mounts in these $Cx3cr1^{CreER}$:$R26^{tomato}$ animals revealed a prominent co-localization of the tomato signal with Iba1 staining, indicating high recombination efficiency in microglia (Appendix Fig S2).

We next generated $Cx3cr1^{CreER}$:$Ifnar1^{fl/fl}$ mice that did not show any apparent phenotypic abnormalities. $Cx3cr1^{CreER}$ mice and $Ifnar1^{fl/fl}$ mice carrying the floxed $Ifnar$ alleles but lacking Cre expression were used as controls. PCR-based genotyping and Western blotting with a specific antibody confirmed Cre-mediated

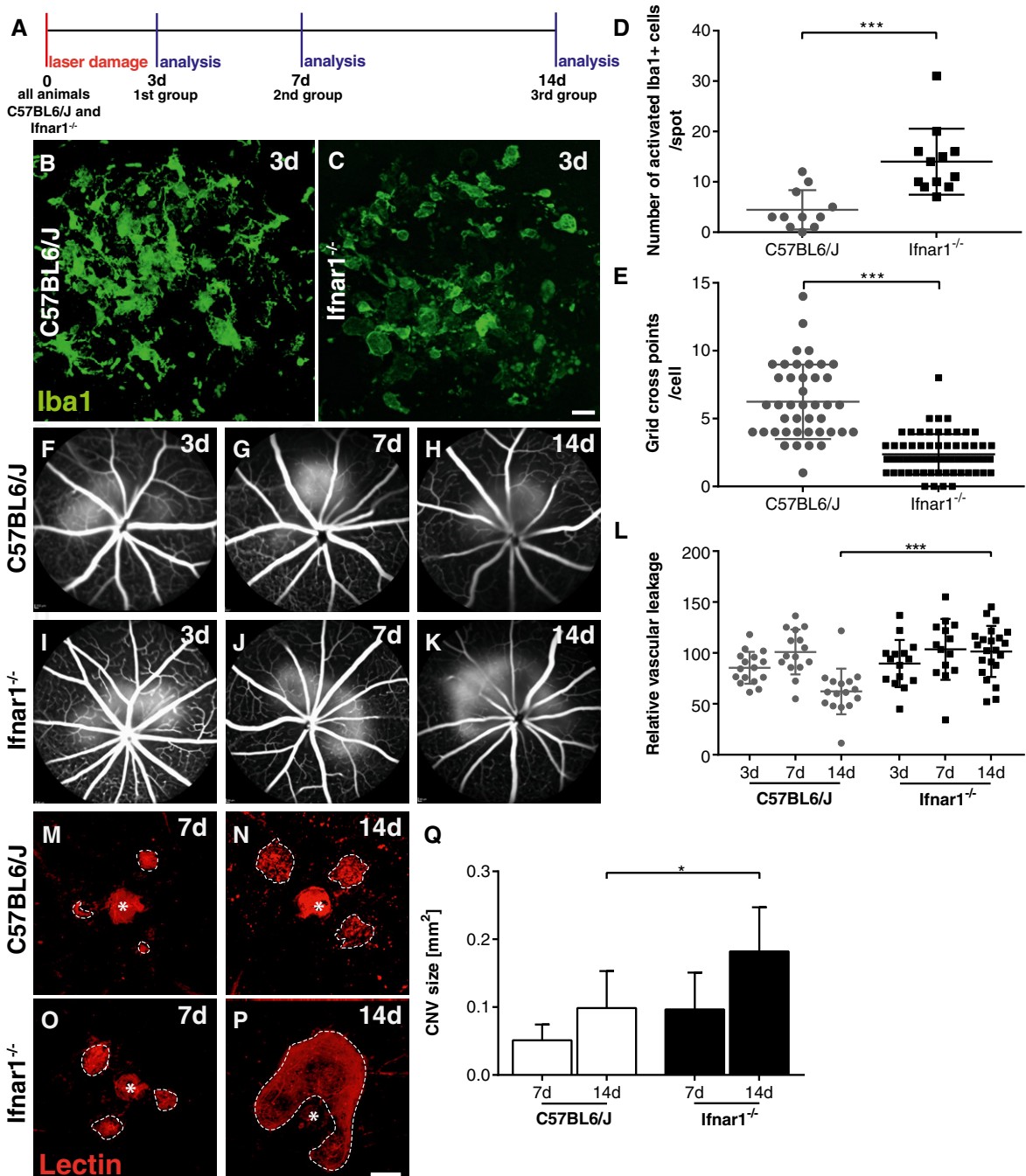

**Figure 1.  Loss of Ifnar1 signaling elicits inflammatory processes and angiogenesis in laser-induced retinal damage.**

A       Experimental design. Laser coagulation was performed in C57BL6/J control and *Ifnar1⁻/⁻* mice. Animals were analyzed 3, 7, and 14 days after laser treatment.

B, C    Representative Iba1 stainings of retinal flat mounts detecting microglia/macrophages in laser spots 3 days after laser coagulation in C57BL6/J controls (B) and *Ifnar1⁻/⁻* (C) mice. Scale bar: 20 μm.

D       Quantification of amoeboid-shaped mononuclear phagocytes in laser spots. Values show mean ± SD (*n* = 11–12 retinas; unpaired Student's *t*-test: ***P = 0.0004).

E       Quantification of immune cell morphology in laser spots using a grid image analysis system. Values show mean ± SD (*n* = 42–62 cells; unpaired Student's *t*-test: ***P < 0.0001).

F–K     Representative fundus fluorescein angiography images of C57BL6/J (F–H) and *Ifnar1⁻/⁻* (I–K) mice 3, 7, and 14 days after laser-induced damage.

L       Quantification of vascular leakage by analyzing pixel intensities at 3, 7, and 14 days after laser-induced retinal damage in C57BL6/J controls versus *Ifnar1⁻/⁻* mice. Values show mean ± SD (*n* = 14–22 eyes; one-way ANOVA followed by Tukey's post-test: ***P < 0.0001).

M–P     Representative images of lectin-stained choroidal flat mounts 7 and 14 days after laser coagulation in C57BL6/J control mice (M, N) and *Ifnar1⁻/⁻* animals (O, P). Dashed lines indicate CNV areas, and the asterisk marks the central optic nerve head. Scale bar: 200 μm.

Q       Quantification of lectin-stained CNV areas with ImageJ software. Bars show mean ± SD (*n* = 4–11 RPE/choroidal flat mounts; one-way ANOVA followed by Tukey's post-test: *P = 0.0281).

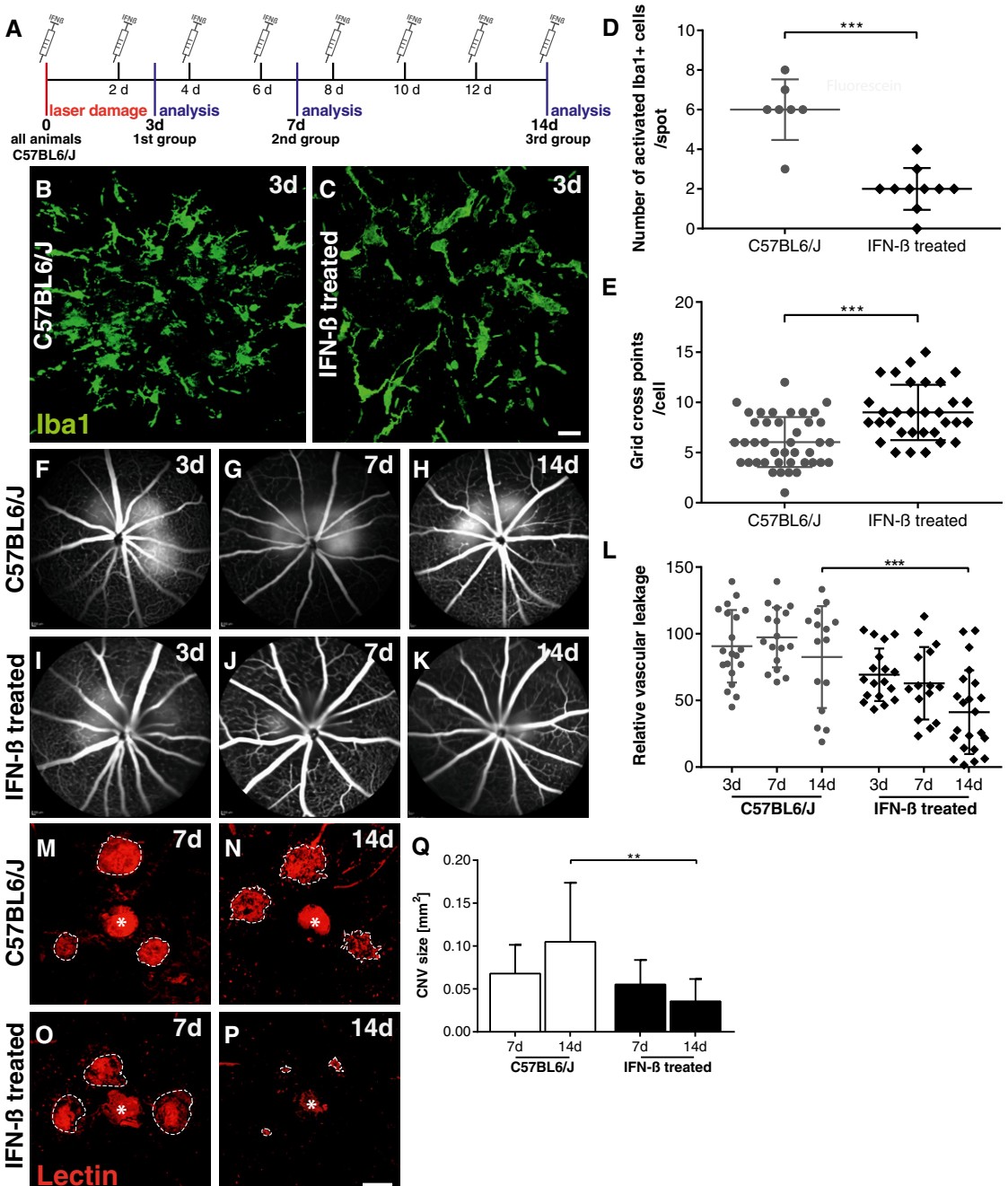

**Figure 2. IFN-β ameliorates microgliosis and inhibits choroidal neovascularization.**

A Experimental design. Laser coagulation was performed in all C57BL6/J mice that were either untreated or received 10,000 units IFN-β i.p. every second day until animals were analyzed 3, 7, and 14 days after laser treatment, respectively.

B, C Representative Iba1 stainings of retinal flat mounts detecting microglia/macrophages in laser spots 3 days after laser coagulation in control mice (B) or IFN-β-treated animals (C). Scale bar: 20 μm.

D Quantification of amoeboid-shaped mononuclear phagocytes in laser spots. Values show mean ± SD ($n$ = 7–10 retinas; unpaired Student's $t$-test: ***$P$ < 0.0001).

E Quantification of immune cell morphology in laser spots using a grid image analysis system. Values show mean ± SD ($n$ = 41–62 cells; unpaired Student's $t$-test: ***$P$ < 0.0001).

F–K Representative fundus fluorescein angiography images of control mice (F–H) or IFN-β-treated animals (I–K) 3, 7, and 14 days after laser-induced damage.

L Quantification of vascular leakage by analyzing pixel intensities at 3, 7, and 14 days after laser-induced retinal damage in control mice and IFN-β-treated animals. Values show mean ± SD ($n$ = 15–22 eyes; one-way ANOVA followed by Tukey's post-test: ***$P$ = 0.0004).

M–P Representative images of lectin-stained choroidal flat mounts 7 and 14 days after laser coagulation in control mice (M, N) and IFN-β-treated animals (O, P). Dashed lines indicate CNV areas, and the asterisk marks the central optic nerve head. Scale bar: 200 μm.

Q Quantification of lectin-stained CNV areas with ImageJ software. Bars show mean ± SD ($n$ = 7–12 RPE/choroidal flat mounts; one-way ANOVA followed by Tukey's post-test: **$P$ = 0.0038).

**Figure 3.  Loss of Ifnar1 signaling in mononuclear phagocytes enhances CNV.**

A    Experimental design. Laser coagulation was performed in tamoxifen-treated *Cx3cr1*^CreER^, *Ifnar1*^fl/fl^, and *Cx3cr1*^CreER^:*Ifnar1*^fl/fl^ mice. Animals were analyzed 3, 7, and 14 days after laser treatment.

B–D    Representative Iba1 staining results of retinal flat mounts detecting microglia/macrophages in laser spot 3 days after laser coagulation in *Cx3cr1*^CreER^ (B), *Ifnar1*^fl/fl^ (C), and *Cx3cr1*^CreER^:*Ifnar1*^fl/fl^ (D) mice. Scale bar: 20 μm.

E    Quantification of amoeboid-shaped mononuclear phagocytes in laser spots. Values show mean ± SD ($n$ = 17–29 retinas; unpaired Student's $t$-test: ***$P$ = 0.0003, **$P$ = 0.0044).

F    Quantification of immune cell morphology in laser spots using a grid image analysis system. Values show mean ± SD ($n$ = 44–52 cells; unpaired Student's $t$-test: ***$P$ < 0.0001).

G–O    Representative fundus fluorescein angiography images of *Cx3cr1*^CreER^ (G–I), *Ifnar1*^fl/fl^ (J–L), and *Cx3cr1*^CreER^:*Ifnar1*^fl/fl^ (M–O) mice 3, 7, and 14 days after laser-induced damage.

P    Quantification of vascular leakage by analyzing pixel intensities at 3, 7, and 14 days after laser-induced retinal damage in *Cx3cr1*^CreER^, *Ifnar1*^fl/fl^, and *Cx3cr1*^CreER^: *Ifnar1*^fl/fl^ mice. Values show mean ± SD ($n$ = 5–12 eyes; one-way ANOVA followed by Tukey's post-test: **$P$ = 0.0032, *$P$ = 0.0247).

Q–V    Representative images of lectin-stained choroidal flat mounts 7 and 14 days after laser coagulation in *Cx3cr1*^CreER^ (Q, R), *Ifnar1*^fl/fl^ (S, T), and *Cx3cr1*^CreER^:*Ifnar1*^fl/fl^ (U, V) mice. Dashed lines indicate CNV areas, and the asterisk marks the central optic nerve head. Scale bar: 200 μm.

W    Quantification of lectin-stained CNV areas with ImageJ software. Bars show mean ± SD ($n$ = 4–11 RPE/choroidal flat mounts; one-way ANOVA followed by Tukey's post-test: *$P$ = 0.043, ***$P$ = 0.0007).

excision of *Ifnar1* exon 10 (Appendix Fig S3A) and reduced Ifnar1 expression (Appendix Fig S3B) in retinal extracts of tamoxifen-injected *Cx3cr1*^CreER^:*Ifnar1*^fl/fl^ mice, respectively. Co-staining of retinal sections with Ifnar1 and Iba1 also revealed only weak Ifnar1 signals that were mainly confined to Iba1^+^ monocytes located in inner retinal blood vessels (Appendix Fig S3C).

All three mouse lines were then subjected to laser coagulation and further analysis of immune cell behavior and CNV formation (Fig 3A). Ifnar-deficient retinal Iba1^+^ cells displayed a strong amoeboid morphology compared to lesion-associated cells from *Cx3cr1*^CreER^ and *Ifnar1*^fl/fl^ mice (Fig 3B–E). The mononuclear phagocytes in *Cx3cr1*^CreER^:*Ifnar1*^fl/fl^ animals also showed significantly less ramifications in the grid cross analysis system, indicating a more reactive phenotype (Fig 3F). We then analyzed vascular leakage and CNV formation in tamoxifen-treated *Cx3cr1*^CreER^:*Ifnar1*^fl/fl^ mice and their two control counterparts. Both control mouse lines developed a typical laser-induced vascular leakage at all three time points (Fig 3G–L). In contrast, *Cx3cr1*^CreER^:*Ifnar1*^fl/fl^ animals showed a significantly increased vascular leakage after 14 days (Fig 3M–P). Quantification of lectin-stained flat mounts then also revealed a significantly larger CNV size at 14 days after laser treatment in *Cx3cr1*^CreER^:*Ifnar1*^fl/fl^ animals compared to *Cx3cr1*^CreER^ and *Ifnar1*^fl/fl^ controls (Fig 3Q–W). Finally, we analyzed the temporal correlation of Iba1^+^ cells with CNV lesions in retinal sections at day 3 and RPE/choroidal flat mounts at day 7 in all three mouse lines (Fig EV3). There was a clear overlap of amoeboid Iba1^+^ cells with increased lectin staining in tamoxifen-treated *Cx3cr1*^CreER^:*Ifnar1*^fl/fl^ animals compared to *Cx3cr1*^CreER^ and *Ifnar1*^fl/fl^ controls at both time points (Fig EV3). These data together clearly demonstrate that loss of Ifnar signaling in mononuclear phagocytes only causes sustained immune cell activation and exacerbates the development of laser-induced CNV lesions.

## Discussion

Our study demonstrates a strong influence of Ifnar1 signaling on retinal microglia and macrophage activity and angiogenesis in the murine laser-induced photocoagulation model that mimics several features of neovascular AMD (Lambert *et al*, 2013). *Ifnar1*^−/−^ animals, lacking the α-subunit of the type 1 interferon α/β-receptor (Muller *et al*, 1994), showed a stronger and prolonged disease

course after laser induction that was associated with an accumulation of predominantly amoeboid mononuclear phagocytes at the lesion sites. These results are in full agreement with a previous study reporting that interferon-β/Ifnar1 signaling has potent immunomodulatory effects in experimental autoimmune encephalomyelitis (EAE), a model for autoimmune brain inflammation and multiple sclerosis (Prinz *et al*, 2008). IFN-β was produced locally in the CNS of EAE animals, and mice lacking Ifnar1 expression in all tissues developed exacerbated clinical disease symptoms accompanied by stronger inflammation, demyelination, and lethality (Prinz *et al*, 2008). This negative influence of Ifnar1 deficiency on EAE was not seen upon Cre-mediated deletion of *Ifnar* in B cell or T cells but was fully recapitulated by LysM-Cre-directed deletion of *Ifnar* in myeloid cells (Prinz *et al*, 2008). Similar findings were reported for the Ifnar1 ligand IFN-β, where its gene disruption in mice caused augmented and sustained demyelination in EAE (Teige *et al*, 2003).

Microglia evolve from distinct primitive yolk sac progenitors (Kierdorf *et al*, 2013) and can be regarded as the primary innate immune effector cells in pathologies of the brain and the retina (Nimmerjahn *et al*, 2005; Kettenmann *et al*, 2011; Karlstetter *et al*, 2015; Zhao *et al*, 2015). To specifically address the role of Ifnar1 in retinal microglia function, we made use of a tamoxifen-inducible *Cx3cr1*^CreER^ mouse. This mouse line was established to specifically target microglia *in vivo*, by facilitating inducible microglia-specific gene deletion in adult animals (Goldmann *et al*, 2013; Yona *et al*, 2013). Despite the redundant expression of Cx3cr1 on all myeloid cell populations, microglia can be distinguished from CNS infiltrating monocytes in *Cx3cr1*^CreER^ mice by their unique features of self-renewal and longevity (Wieghofer *et al*, 2015). However, other long-lived resident mononuclear phagocyte populations located close to the retina such as choroidal macrophages may be also potentially targeted by this system (McMenamin, 1999). Our laser-CNV analysis in tamoxifen-induced *Cx3cr1*^CreERT2^:*Ifnar1*^fl/fl^ animals showed a significantly enhanced retinal pathology comparable to that of complete Ifnar1 deletion. Thus, the disease-promoting effects of Ifnar1 deletion seem to be confined to microglia and potentially other long-lived macrophage subsets, revealing a significant contribution of these cell types to increase angiogenesis in the laser-CNV model.

Since IFN-β therapy is an established treatment option in relapsing remitting multiple sclerosis (Steinman *et al*, 2012) and inhibits

   

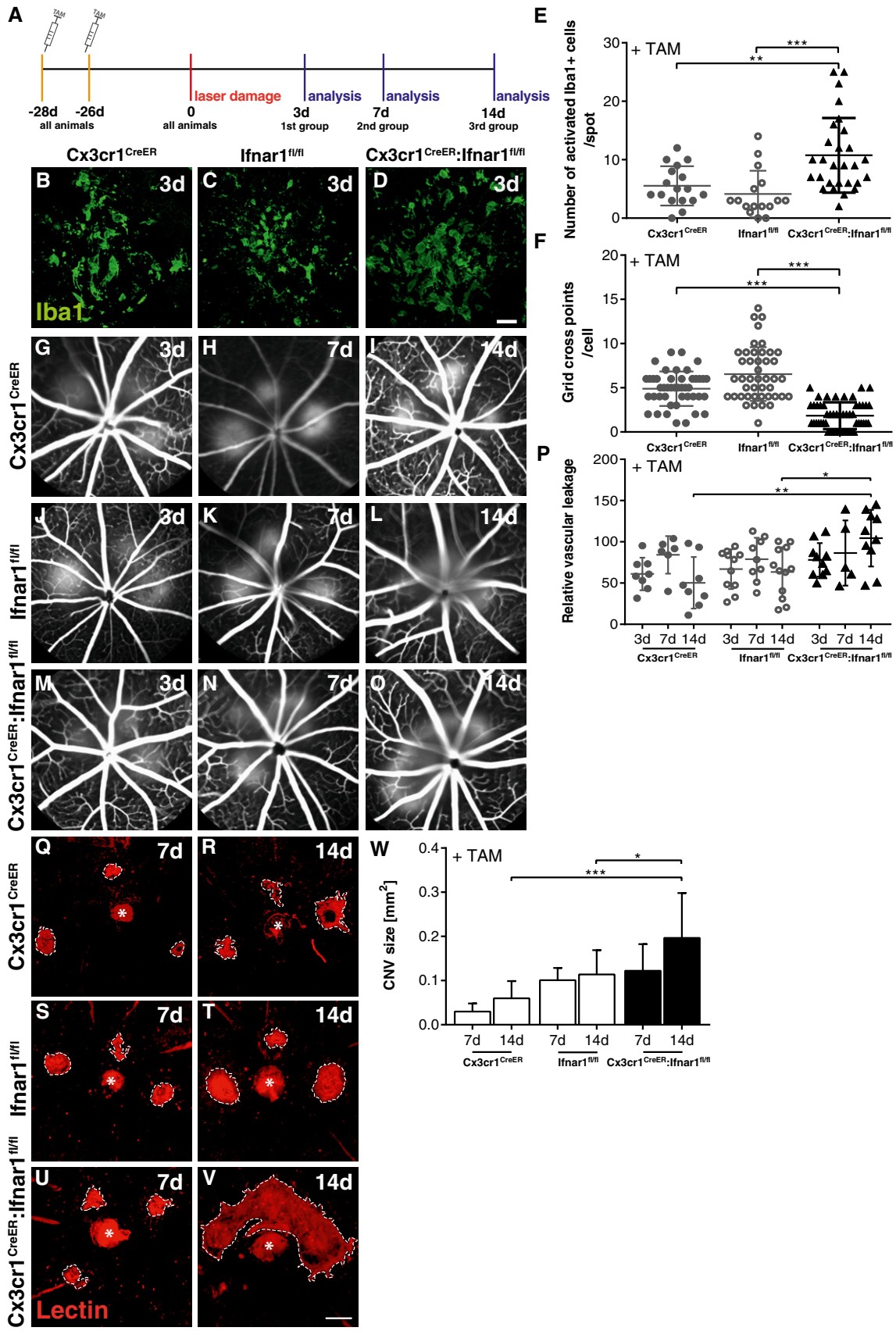

**Figure 3.**

EAE via different cellular and humoral mechanisms (Inoue & Shinohara, 2013), we postulated an immunomodulatory potential in the eye. Our data clearly revealed decreased microgliosis and less CNV in the laser-damage model. This is in agreement with data that showed protective effects of IFN-β in experimental autoimmune uveoretinitis, a model for human intraocular inflammation, by suppressing Th1 and Th17 cells (Sun *et al*, 2011). Moreover, systemic IFN-β was tested in a CNV rabbit model without directly analyzing microglia/macrophage reactions (Yasukawa *et al*, 2002). In this model, subretinal injection of gelatin microspheres containing basic fibroblast growth factor (bFGF) triggered neovascularization for approximately 4 weeks. Continuous systemic therapy with dextran-conjugated IFN-β was very effective in lowering CNV in the incipient stage but did not affect CNV progression in later phases (Yasukawa *et al*, 2002). In contrast, our data showed significant effects of IFN-β especially in the late phase. We hypothesize that the laser-coagulation model mainly involves chronic inflammatory events whereas the bFGF model may act predominantly via the formation of neovascular membrane scars. In accordance with this hypothesis, IFN-β treatment also ameliorated laser-induced CNV in rabbits (Kimoto *et al*, 2002) and monkeys (Tobe *et al*, 1995). Of note, a patient with multiple sclerosis and punctate inner choroidopathy could significantly profit from systemic IFN-β therapy and was subsequently free of active CNV (Cirino *et al*, 2006).

Taken together, our data in the laser-coagulation model showed that Ifnar1 deficiency enhanced retinal microglia/macrophage reactivity and that IFN-β inhibited this immune cell activation, vessel leakage, and CNV. Targeting Ifnar1/IFN-β signaling may therefore highlight new therapeutic strategies for AMD and potentially other chronic inflammatory and degenerative diseases of the retina.

# Materials and Methods

### Animals and tamoxifen administration

Experiments were conducted with 6- to 10-week-old male and female *Ifnar1*$^{-/-}$ mice (Muller *et al*, 1994) and *Cx3cr1*$^{CreER}$:*Ifnar1*$^{fl/fl}$ mice, which were obtained by breeding *Cx3cr1*$^{CreER}$ mice (Yona *et al*, 2013) and *Ifnar1*$^{fl/fl}$ animals (Kamphuis *et al*, 2006; Detje *et al*, 2009). *Cx3cr1*$^{CreER}$ mice were crossed with *R26*$^{tomato}$ reporter mice (Soriano, 1999). *Ifnar1*$^{-/-}$, *Ifnar1*$^{fl/fl}$, and *R26*$^{tomato}$ mice were on C57BL6/J and *Cx3cr1*$^{CreER}$ mice were on C57BL6/N background. All animals were maintained in an air-conditioned environment at 22°C on a 12-h light–dark schedule, had access to phytoestrogen-free food and water *ad libitum*, and were health-monitored on a regular basis. For induction of Cre recombinase, *Cx3cr1*$^{CreER}$ mice and *Cx3cr1*$^{CreER}$:*Ifnar1*$^{fl/fl}$ mice were treated with 4 mg tamoxifen (T5648; Sigma) dissolved in 200 μl corn oil (C8267; Sigma) injected subcutaneously at two time points 48 h apart. The animals had consecutive numbers which were allocated to the genotype only after complete experimental evaluation. All experiments were approved by the governmental body responsible for animal welfare in the state of North Rhine-Westphalia, Germany (Landesamt für Natur, Umwelt und Verbraucherschutz Nordrhein-Westfalen, Germany), with the permission number Az 84-02-04-2014-A466.

### Laser coagulation

Laser damage of the retina was performed using a slit-lamp-mounted diode laser system (Viridis). The mice were anesthetized by an intraperitoneal injection of 150 μl ketamine hydrochloride (100 mg/kg body weight, Ketavet; Pfizer Animal Health) and xylazine hydrochloride (5 mg/kg body weight, 2% Rompun; Bayer HealthCare) diluted in 0.9% sodium chloride. Before laser treatment, the pupils of the animals were dilated using Phenylephrin 2.5%–Tropicamid 0.5%. Three laser burns were created equally distributed around the optic nerve of both eyes (energy 125 mW, duration 100 ms, spot size 100 μm). The animals were divided into three groups and analyzed after 3, 7, or 14 days.

### Interferon-β administration

Animals were treated with recombinant human interferon-beta 1a, produced in CHO cells (300-02BC; PeproTech). A dose of 10,000 U diluted in 100 μl phosphate-buffered saline (PBS) was injected i.p. every other day from day 0 until the end of the experiment. To minimize the effect of subjective bias, mice were treated cage-wise and cages were allocated randomly to the experimental groups.

### Preparation of flat mounts and immunohistochemistry

The eyes were enucleated and fixed in 4% paraformaldehyde (0335.2; Roth) for 4 h at room temperature. Retinal and RPE/choroidal flat mounts were dissected and retinal flat mounts were permeabilized overnight (5% Triton X-100, 5% Tween-20 in PBS). After blocking the unspecific antigens with BLOTTO (1% milk powder, 0.01% Triton X-100 in PBS) for 1 h at room temperature, the retinal flat mounts were incubated in the primary antibody overnight at 4°C (Iba1, rabbit polyclonal, 019-19741; Wako Chemicals). The RPE/choroidal flat mounts were incubated for permeabilization and blocking in 1% BSA (A2153; Sigma), 5% nonspecific goat serum (16210-064; Gibco), and 0.3% Triton X-100 in PBS overnight at 4°C, followed by an incubation with a 1:500 dilution of Iba1 antibody and 1:10 dilution of primary TRITC-conjugated lectin (L5264; Sigma) to label microglia/macrophages and endothelial cells, respectively. All flat mounts were washed in PBS before incubating with a 1:1,000 dilution of goat anti-rabbit AlexaFluor 488 nm-conjugated secondary antibody (A11008; Life Technologies). After washing, retinal and RPE/choroidal flat mounts were mounted on a microscope slide and embedded with fluorescence mounting medium (S3023; DakoCytomation).

### Image analysis

The stainings were evaluated with a Zeiss Imager M.2 with an ApoTome.2. The total number of Iba1$^+$ cells and the number of amoeboid-shaped cells were counted within a circular region of 200 μm diameter around the laser spots. Cellular morphology was analyzed using a grid system to determine the number of grid crossing points per cell ($n$ = 40–70 cells; from at least 3 different retinas per group) (Chen *et al*, 2012). CNV areas in RPE/choroidal flat mounts were quantified with the spline function of the graphic tool of ZEN software (Zeiss).

    

## Fundus fluorescein angiography (FFA)

The animals were anesthetized, the pupils were dilated, and the vascular leakage was determined with FFA using the Spectralis™ HRA device. Following SD-OCT analysis, 100 µl of 2.5% fluorescein (Alcon) diluted in 0.9% sodium chloride was injected intraperitoneally. For FFA analysis, late-phase pictures (10 min after fluorescein injection) were taken. To quantify the laser-induced vascular leakage, the pixel intensity was measured in two regions of interests (ROIs) within and one ROI outside each laser spot using the image processing program ImageJ (NIH). Because three laser spots were induced per eye, we quantified the pixel intensity of six ROIs within and three ROIs outside the fluorescein leakages. After averaging the values and subtracting the background, one data point represented the mean laser-induced leakage per eye. Eyes were excluded from the analysis in case of choroidal hemorrhages and when laser lesions completely fused with each other or the optic nerve head.

## Statistical analysis

Statistical analysis was performed using GraphPad Prism 6. For analysis of two groups, unpaired Student's *t*-test was performed. To compare more than two groups, one-way ANOVA followed by Tukey's post-test was used. All values are presented as mean ± SD. $P$-values ≤ 0.05 were considered significant.

Expanded View for this article is available online.

## Acknowledgements

The authors thank Prof. Thomas Wunderlich, Max Planck Institute for Metabolism Research, and Institute for Genetics, University of Cologne, for providing *R26*[tomato] mice. We also thank Prof. Andrew Dick, Unit of Ophthalmology, School of Clinical Sciences and School of Cellular and Molecular Medicine, University of Bristol, for helpful discussion and Khalid Rashid for critical reading of the manuscript. This work was supported by grants from the Graduate Program in Pharmacology and Experimental Therapeutics at the University of Cologne in collaboration with Bayer AG, the DFG (LA1203/6-2, LA1203/9-1, LA1203/10-1, and FOR2240), the ProRetina Foundation, and the Hans and Marlies Stock Foundation.

## Author contributions

AL performed experiments, analyzed data, and wrote the manuscript. AC performed laser damage and angiography. RS performed experiments, and MP and UK contributed *Ifnar1*[−/−] and *Cx3cr1*[CreER]:*Ifnar1*[fl/fl] mice and corrected the manuscript. TL designed and supervised the study, obtained funding, and finalized the manuscript. All authors read and contributed to the final manuscript.

## Conflict of interest

The authors declare that they have no conflict of interest.

## References

Augood CA, Vingerling JR, de Jong PT, Chakravarthy U, Seland J, Soubrane G, Tomazzoli L, Topouzis F, Bentham G, Rahu M *et al* (2006) Prevalence of age-related maculopathy in older Europeans: the European Eye Study (EUREYE). *Arch Ophthalmol* 124: 529–535

### The paper explained

#### Problem

Age-related macular degeneration (AMD) is a leading cause of blindness in the Western World with increasing prevalence in the aging population. AMD is often accompanied by an activation of the innate immune system involving complement factors and reactive mononuclear phagocytes. Interferon-β is an immunomodulatory drug to treat multiple sclerosis, an autoimmune demyelinating disease of the brain. It was previously unknown whether interferon-β signaling in retinal immune cells may be a therapy target for retinal degenerative diseases including AMD.

#### Results

Using the laser-coagulation model of neovascular AMD in complete or microglia/macrophage-specific Ifnar1-deficient mice, we report an essential role of interferon-β signaling in regulating immune cell reactivity and pathological angiogenesis. Loss of Ifnar1 triggered microglia/macrophage activation, vessel leakage, and choroidal neovascularization (CNV). In contrast, IFN-β therapy attenuated retinal immune cell response and CNV development.

#### Impact

Our findings indicate a key role for Ifnar1 signaling in retinal immune mechanisms. Although studied in an experimental model of laser-induced CNV in mice, the immunomodulatory potential of IFN-β is a promising new strategy for future therapy approaches to control chronic inflammation in AMD.

Axtell RC, Steinman L (2008) Type 1 interferons cool the inflamed brain. *Immunity* 28: 600–602

Bruttger J, Karram K, Wortge S, Regen T, Marini F, Hoppmann N, Klein M, Blank T, Yona S, Wolf Y *et al* (2015) Genetic cell ablation reveals clusters of local self-renewing microglia in the Mammalian Central Nervous System. *Immunity* 43: 92–106

Chen M, Zhao J, Luo C, Pandi SP, Penalva RG, Fitzgerald DC, Xu H (2012) Para-inflammation-mediated retinal recruitment of bone marrow-derived myeloid cells following whole-body irradiation is CCL2 dependent. *Glia* 60: 833–842

Cirino AC, Mathura JR Jr, Jampol LM (2006) Resolution of activity (choroiditis and choroidal neovascularization) of chronic recurrent punctate inner choroidopathy after treatment with interferon B-1A. *Retina* 26: 1091–1092

Combadière C, Feumi C, Raoul W, Keller N, Rodéro M, Pézard A, Lavalette S, Houssier M, Jonet L, Picard E *et al* (2007) CX3CR1-dependent subretinal microglia cell accumulation is associated with cardinal features of age-related macular degeneration. *J Clin Invest* 117: 2920–2928

Damani MR, Zhao L, Fontainhas AM, Amaral J, Fariss RN, Wong WT (2011) Age-related alterations in the dynamic behavior of microglia. *Aging Cell* 10: 263–276

Dann A, Poeck H, Croxford AL, Gaupp S, Kierdorf K, Knust M, Pfeifer D, Maihoefer C, Endres S, Kalinke U *et al* (2011) Cytosolic RIG-I-like helicases act as negative regulators of sterile inflammation in the CNS. *Nat Neurosci* 15:98–106

Detje CN, Meyer T, Schmidt H, Kreuz D, Rose JK, Bechmann I, Prinz M, Kalinke U (2009) Local type I IFN receptor signaling protects against virus spread within the central nervous system. *J Immunol* 182: 2297–2304

Goldmann T, Wieghofer P, Muller PF, Wolf Y, Varol D, Yona S, Brendecke SM, Kierdorf K, Staszewski O, Datta M *et al* (2013) A new type of microglia

gene targeting shows TAK1 to be pivotal in CNS autoimmune inflammation. *Nat Neurosci* 16: 1618−1626

Gupta N, Brown KE, Milam AH (2003) Activated microglia in human retinitis pigmentosa, late-onset retinal degeneration, and age-related macular degeneration. *Exp Eye Res* 76: 463−471

Inoue M, Shinohara ML (2013) The role of interferon-β in the treatment of multiple sclerosis and experimental autoimmune encephalomyelitis - in the perspective of inflammasomes. *Immunology* 139: 11−18

Jager RD, Mieler WF, Miller JW (2008) Age-related macular degeneration. *N Engl J Med* 358: 2606−2617

Jin S, Kawanokuchi J, Mizuno T, Wang J, Sonobe Y, Takeuchi H, Suzumura A (2007) Interferon-beta is neuroprotective against the toxicity induced by activated microglia. *Brain Res* 1179: 140−146

Kamphuis E, Junt T, Waibler Z, Forster R, Kalinke U (2006) Type I interferons directly regulate lymphocyte recirculation and cause transient blood lymphopenia. *Blood* 108: 3253−3261

Karlstetter M, Scholz R, Rutar M, Wong WT, Provis JM, Langmann T (2015) Retinal microglia: just bystander or target for therapy? *Prog Retin Eye Res* 45: 30−57

Kettenmann H, Hanisch UK, Noda M, Verkhratsky A (2011) Physiology of microglia. *Physiol Rev* 91: 461−553

Kierdorf K, Erny D, Goldmann T, Sander V, Schulz C, Perdiguero EG, Wieghofer P, Heinrich A, Riemke P, Holscher C *et al* (2013) Microglia emerge from erythromyeloid precursors via Pu.1- and Irf8-dependent pathways. *Nat Neurosci* 16: 273−280

Kimoto T, Takahashi K, Tobe T, Fujimoto K, Uyama M, Sone S (2002) Effects of local administration of interferon-beta on proliferation of retinal pigment epithelium in rabbit after laser photocoagulation. *Jpn J Ophthalmol* 46: 160−169

Lambert V, Lecomte J, Hansen S, Blacher S, Gonzalez ML, Struman I, Sounni NE, Rozet E, de Tullio P, Foidart JM *et al* (2013) Laser-induced choroidal neovascularization model to study age-related macular degeneration in mice. *Nat Protoc* 8: 2197−2211

Langmann T (2007) Microglia activation in retinal degeneration. *J Leukoc Biol* 81: 1345−1351

Levy O, Calippe B, Lavalette S, Hu SJ, Raoul W, Dominguez E, Housset M, Paques M, Sahel JA, Bemelmans AP *et al* (2015) Apolipoprotein E promotes subretinal mononuclear phagocyte survival and chronic inflammation in age-related macular degeneration. *EMBO Mol Med* 7: 211−226

McMenamin PG (1999) Dendritic cells and macrophages in the uveal tract of the normal mouse eye. *Br J Ophthalmol* 83: 598−604

Muller U, Steinhoff U, Reis LF, Hemmi S, Pavlovic J, Zinkernagel RM, Aguet M (1994) Functional role of type I and type II interferons in antiviral defense. *Science* 264: 1918−1921

Nimmerjahn A, Kirchhoff F, Helmchen F (2005) Resting microglial cells are highly dynamic surveillants of brain parenchyma in vivo. *Science* 308: 1314−1318

Noel A, Jost M, Lambert V, Lecomte J, Rakic JM (2007) Anti-angiogenic therapy of exudative age-related macular degeneration: current progress and emerging concepts. *Trends Mol Med* 13: 345−352

Prinz M, Schmidt H, Mildner A, Knobeloch KP, Hanisch UK, Raasch J, Merkler D, Detje C, Gutcher I, Mages J *et al* (2008) Distinct and nonredundant

in vivo functions of IFNAR on myeloid cells limit autoimmunity in the central nervous system. *Immunity* 28: 675−686

Rofagha S, Bhisitkul RB, Boyer DS, Sadda SR, Zhang K, SEVEN-UP Study Group (2013) Seven-year outcomes in ranibizumab-treated patients in ANCHOR, MARINA, and HORIZON: a multicenter cohort study (SEVEN-UP). *Ophthalmology* 120: 2292−2299

Sadler AJ, Williams BR (2008) Interferon-inducible antiviral effectors. *Nat Rev Immunol* 8: 559−568

Sene A, Chin-Yee D, Apte RS (2015) Seeing through VEGF: innate and adaptive immunity in pathological angiogenesis in the eye. *Trends Mol Med* 21: 43−51

Sierra A, Gottfried-Blackmore AC, McEwen BS, Bulloch K (2007) Microglia derived from aging mice exhibit an altered inflammatory profile. *Glia* 55: 412−424

Soriano P (1999) Generalized lacZ expression with the ROSA26 Cre reporter strain. *Nat Genet* 21: 70−71

Steinman L, Merrill JT, McInnes IB, Peakman M (2012) Optimization of current and future therapy for autoimmune diseases. *Nat Med* 18: 59−65

Sun M, Yang Y, Yang P, Lei B, Du L, Kijlstra A (2011) Regulatory effects of IFN-beta on the development of experimental autoimmune uveoretinitis in B10RIII mice. *PLoS One* 6: e19870

Teige I, Treschow A, Teige A, Mattsson R, Navikas V, Leanderson T, Holmdahl R, Issazadeh-Navikas S (2003) IFN-beta gene deletion leads to augmented and chronic demyelinating experimental autoimmune encephalomyelitis. *J Immunol* 170: 4776−4784

Tobe T, Takahashi K, Ohkuma H, Uyama M (1995) The effect of interferon-beta on experimental choroidal neovascularization. *Nippon Ganka Gakkai Zasshi* 99: 571−581

Touil T, Fitzgerald D, Zhang GX, Rostami A, Gran B (2006) Cutting Edge: TLR3 stimulation suppresses experimental autoimmune encephalomyelitis by inducing endogenous IFN-beta. *J Immunol* 177: 7505−7509

Wieghofer P, Knobeloch KP, Prinz M (2015) Genetic targeting of microglia. *Glia* 63: 1−22

Yasukawa T, Kimura H, Tabata Y, Kamizuru H, Miyamoto H, Honda Y, Ogura Y (2002) Targeting of interferon to choroidal neovascularization by use of dextran and metal coordination. *Invest Ophthalmol Vis Sci* 43: 842−848

Yona S, Kim KW, Wolf Y, Mildner A, Varol D, Breker M, Strauss-Ayali D, Viukov S, Guilliams M, Misharin A *et al* (2013) Fate mapping reveals origins and dynamics of monocytes and tissue macrophages under homeostasis. *Immunity* 38: 79−91

Zhao L, Zabel MK, Wang X, Ma W, Shah P, Fariss RN, Qian H, Parkhurst CN, Gan WB, Wong WT (2015) Microglial phagocytosis of living photoreceptors contributes to inherited retinal degeneration. *EMBO Mol Med* 7: 1179−1197

