## [Review Process File · EMBO Molecular Medicine]

Interferon-beta signaling in retinal mononuclear phagocytes attenuates pathological neovascularization

Anika Lückoff, Albert Caramoy, Rebecca Scholz, Marco Prinz, Ulrich Kalinke, and Thomas Langmann

Corresponding author: Thomas Langmann, University of Cologne

Review timeline:

Submission date:	26 October 2015
Editorial Decision:	23 November 2015
Revision received:	10 March 2016
Editorial Decision:	30 March 2016
Revision received:	01 April 2016
Accepted:	06 April 2016

Transaction Report:

Editor: Céline Carret

1st Editorial Decision

23 November 2015

Thank you for the submission of your manuscript to EMBO Molecular Medicine. We have now heard back from the three referees whom we asked to evaluate your manuscript. Although the referees find the study to be of potential interest, they also raise a number of concerns that must be addressed in the next final version of your article.

You will see from the comments below that all referees find the study interesting and referee 3 particularly highlights its translational relevance. While referees 2 and 3 are generally more supportive, referee 1 is rather critical of the experimental design and is concerned about the CNV model and the mouse model used as well as existing literature previously establishing that IFN β inhibits laser-induced CNV in rabbits and monkeys. As we feel that requested experiments would certainly improve the robustness of the data, we would like to invite you to address all issues as suggested, and experimentally when needed.

Given these evaluations, I would like to give you the opportunity to revise your manuscript, with the understanding that the referee concerns must be fully addressed and that acceptance of the manuscript would entail a second round of review. Please note that it is EMBO Molecular Medicine policy to allow only a single round of revision and that, as acceptance or rejection of the manuscript will depend on another round of review, your responses should be as complete as possible.

I look forward to seeing a revised form of your manuscript as soon as possible.

***** Reviewer's comments *****

Referee #1 (Comments on Novelty/Model System):

Generally the laser lesions are (i) too close to the optic nerve (they are classically applied 4-5 optic nerve diameters away from the optic nerve) and (ii) not evenly distributed around the optic nerve. As a result some of the CNV formations fuse between two laser lesions which greatly influences their size as the bridging CNV lesions are bigger than two single CNV lesions.

Referee #1 (Remarks):

This is a potentially interesting manuscript about the role of IFN beta and its receptor IFNAR in subretinal inflammation and choroidal neovascularization. There are however several major problems that should be addressed. Generally the laser lesions are (i) too close to the optic nerve (they are classically applied 4-5 optic nerve diameters away from the optic nerve) and (ii) not evenly distributed around the optic nerve. As a result some of the CNV formations fuse between two laser lesions which greatly influences their size as the bridging CNV lesions are bigger than two single CNV lesions. The inflammation was only analyzed at d3, even though the vascular changes are invariably only observed at d14. If the differences in IBA1+cells at d3 were responsible for the vascular phenotype, you would expect the vascular phenotype to be different at d7. It seems therefore likely that the IFNAR1-dependent differences in inflammatory phenotype that are responsible for the vascular changes occur at a later stage, or they should at least be analyzed additionally at a later time point. The analysis of the Cx3cr1CreER:Ifnar1 fl/fl mice does not allow to specifically analyze microglial IFNAR1 expression per se in this model, as the lesion was made in the choroid that contains numerous resident macrophages that likely express cx3cr1 and also have a slow turn-over. Also the deletion of Ifnar1 was not controlled for in the tissue and the different macrophage populations in the hands of the authors and a Cre-expressing control is missing. IFNAR1 expression is not analyzed in the model and other cell types that might express IFNAR1 and mediate the therapeutic effect of IFN beta are therefore not recognized.

Abstract and discussion:

The authors state in several sentences to have analyzed the specific role of microglial IFNAR1. These conclusions can not be drawn from the presented data as there are likely other resident macrophages with a slow turnover and a decent Cx3cr1 expression that participate in the inflammation associated with the laser-injury. (...interferon- signaling in the retina accelerates microgliosis ... enhanced early microglia reactivity in lesion areas ... IFNAR in microglia only)

Introduction :

The previous study by Yasukawa et al on interferon and CNV in rabbits is not mentioned in the introduction. A previous study on laser-induced CNV in monkeys is not cited anywhere in the manuscript (Nippon Ganka Gakkai Zasshi. 1995 May;99(5):571-81. The effect of interferon-beta on experimental choroidal neovascularization. Tobe T1, Takahashi K, Ohkuma H, Uyama M.) There also is an interesting case report about an interferon beta and CNV that is not cited at all (Retina. 2006 Nov-Dec;26(9):1091-2. Resolution of activity (choroiditis and choroidal neovascularization) of chronic recurrent punctate inner choroidopathy after treatment with interferon B-1A. Cirino AC1, Mathura JR Jr, Jampol LM.)

Results :

Figure 1 : In the experiments represented in Figure 1 the authors analyzed IBA1+cell infiltration (d3) and CNV (d3, d7, d14) in wildtype and Ifnar^{-/-} mice. The results presented in figure 1 do not allow a differentiation between microglia, infiltrating inflammatory macrophages or choroidal resident macrophages. The authors need to be more precise in their wording as the cells they count in the lesions likely are a mixture of all three cell types.

The authors counted amoeboid IBA1+cells and crossing points. The total number of IBA1+cells needs also to be shown. Why were the IBA1+ cells only quantified at d3, when the vascular changes are only apparent at d14 ? A later analysis of IBA1 cell numbers and phenotype is necessary if the authors think that their the IBA1 cell phenotype is the reason for the vascular differences.

The CNV lesions are very close to the optic nerve. In panel N and P, CNV bridging two to three laser lesions are visible. When laser lesions are two close to each other a CNV lesion often forms between two laser lesions, which is bigger than the addition of two separate lesions. I would

therefore suggest not to include bridging CNV lesions, as their size is also influenced by the aleatory distance to the neighbouring CNV. Also I am not sure I understand the quantification method of angiography fluorescence by choosing two regions of interest (ROI) within and one ROI outside the laser spots. Are the CNV size quantifications per/Impact? if so, how were confluent CNVs quantified? Why did the authors use lectin as a vascular stain, as it also marks activated IBA1 cells?

Figure 2: In the experiments represented in Figure 2 the authors analyzed IBA1+cell infiltration (d3) and CNV (d3, d7, d14) in mice treated with IFN beta. Again the IBA1 cell number and phenotypes were only analyzed at d3, when the vascular differences only appeared 11 days later. Also the total number of IBA1+cells is again missing. Panel N reveals confluent CNVs. Also the laser spots are sometimes distributed equally around the optic nerve (M) sometimes only to one side of the ON(N). The authors also measured edema formation in this set of experiments. Edema is a fluid accumulation within the tissue (either intracellularly or intercellularly). How do the authors distinguish between edema and infiltrating cells (IBA1+cells and proliferating endothelial cells) in the OCT images?

Figure 3: In Figure 3 the authors analyzed IBA1+cell infiltration (d3) and CNV (d3, d7, d14) in *Ifnar1* fl/fl mice and tamoxifen-treated *Cx3cr1*CreER:*Ifnar1* fl/fl mice. The authors induced Cre expression 28 and 26 days prior to the laser lesion which should permanently delete the *Ifnar1* gene in cells that express high levels of *Cx3cr1*. 26 days after the TAM treatment cells with a high turnover, such as monocytes will again express *Ifnar*. The authors therefore state that the experimental mice only have a lack of *Ifnar* in microglial cells. This is however likely incorrect, as resident macrophages in choroid and ciliary body also have a very slow turnover. They likely participate in the laser-induced inflammation (they are actually closer to the burn than microglial cells) and will likely still lack (at least in part) *Ifnar* expression. It is therefore not possible to decipher the role of *ifnar* in only microglial cells using these mice. The efficiency of *Ifnar* deletion in *Cx3cr1* expressing cells with high and low turnover were not analyzed in the hands of the authors and no data from *Cx3cr1* cre expressing mice is presented. The *Iba1* cells in panel B (*Ifnar1* fl/fl) look very differently to the ones shown in Fig. 1B (wildtype) why is that? The laser spots are again very close to the optic nerve (one CNV in panel N seems to actually merge with the optic nerve). They are again not distributed equally, sometimes being only on one side of the optic nerve, sometimes all around.

Referee #2 (Comments on Novelty/Model System):

Due to lack of animal models that approximate macular degeneration, laser injury may be acceptable.

Referee #2 (Remarks):

The goal of the study was to analyze the role of interferon (IFN)- β and its receptor (IFNAR) in the laser model of retinal injury. The key findings of this study were: (1) Laser-treated *Ifnar1*^{-/-} mice showed enhanced microglia activity resulting in vascular leakage and choroidal neovascularization (CNV). (2) Laser-treated *Cx3cr1*CreER:*Ifnar1* fl/fl mice (ie depletion of IFNAR in microglia only) showed similar effects. (3) IFN- β treated wild-type mice displayed decreased microgliosis, vascular leakage and CNV lesion size. These results demonstrate a strong influence of IFN- β signaling on retinal microglia activity and may offer new therapeutic strategies for acute retinal injuries.

Major comments

1. The authors should show cross sections of the CNV lesions induced laser coagulation as well as cross sections stained with *Iba1*.
2. It would be informative to include the expression level of *Ifnar* in wt, *Ifnar1*^{-/-} and *Cx3cr1*CreER:*Ifnar1* fl/fl mice in the retina using immunoblot and immunohistochemistry
3. Acute laser injury may not have relevance to a chronic degeneration disorder, including age related macular degeneration.

Minor comments:

1. Laser-treated wild-type mice are labeled as C57BL6/J in Figure 1 and as control in Figure 2 - please make it consistent.
2. Figure 2C: the label "+IFN- β " is not necessary as it

Referee #3 (Comments on Novelty/Model System):

This is an exciting study with significant translational potential. The model systems are adequate but some additional controls would be required especially in the studies using tamoxifen induced Cre recombinase expression in microglial cells. Further details are outlined in my comments below.

Referee #3 (Remarks):

This manuscript by L. Ckoff and colleagues focuses on the role of interferon-beta in the regulation of choroidal neovascularisation, the end stage of the relatively common form of blindness wet age-related macular degeneration (AMD). The authors also show that interferon-beta is intimately involved in regulating microglial homeostasis and a lack of IFN-beta can induce microgliosis.

Using mice lacking the interferon receptor, the authors show that microglial reactivity was increased in regions of laser induced CNV (a model of neovascular AMD). The authors showed similar results in floxed mice that had IFNAR deleted specifically in microglial cells.

Strikingly, and what represents an exciting translational finding, the authors showed that systemic IFN-beta therapy in mice post induction of laser CNV showed significantly decreased CNV lesions and that IFN-beta therapy could have utility in the treatment of neovascular AMD patients.

Below are some specific comments for the authors to address

1. The quantification of amoeboid shaped cells in Fig 1B/D will need to be elaborated upon in the methods section as it strikes me as a very subjective way of data analysis.
2. What is the homology of human IFN-beta to mouse IFN-beta and are there differences in bioactivity between the two? Human IFN-beta only has 60% homology to mouse IFN-beta...would there be differences in therapeutic readout if the authors used mouse IFN-beta?
3. Figure 3 should have included a control group of the Cx3cr1CreER mice on their own. It is widely accepted that Cre recombinase can have biological effects and toxicity when expressed in cells and this control would be important to include in the figure. This control would need to be used in all sub-sections of Figure 3. It will markedly strengthen the paper.

1st Revision - authors' response

10 March 2016

Referee #1

This is a potentially interesting manuscript about the role of IFN beta and its receptor IFNAR in subretinal inflammation and choroidal neovascularization.

There are however several major problems that should be addressed.

1. Generally the laser lesions are (i) too close to the optic nerve (they are classically applied 4-5 optic nerve diameters away from the optic nerve) and (ii) not evenly distributed around the optic nerve. As a result some of the CNV formations fuse between two laser lesions which

greatly influences their size as the bridging CNV lesions are bigger than two single CNV lesions.

Response:

When studying laser-coagulation in mice there are several limitations mainly due to the small eye, large lens and the differences in optics of small rodents. We have done our best to ensure that the laser lesions were equally distributed at 3, 9 and 12 o'clock positions. In case the laser spots seen in IR images fused with each other or with the optic disc the whole eye was excluded from analysis. Our protocol is in accordance with several well accepted publications showing laser lesions in rodents and especially in mice with 1-3 optic nerve diameters away from the ON.

The reference we have cited in our manuscript is: Lambert et al. 2013, Laser-induced choroidal neovascularization model to study age-related macular degeneration in mice. *Nat Protoc* 8: 2197-211. Other good examples can be found in Campa et al. *IOVS* 2008, 49:1178 or Horie et al. *SciRep* 2013, 3:3072 or Gammons et al. *IOVS* 2013, 54:6052 or Zhang et al. *PNAS* 2009, 106:6152-7 and several others.

For the reviewer's information, we show here representative infrared images of the retinal fundus of some experimental mice analyzed in our study. You will see that there were no laser lesions fused with each other or the optic disc. It is however possible, especially at late stage in severely affected *Ifnar1^{-/-}* and *Cx3cr1^{CreER}:Ifnar1^{flox/flox}* mice that sometimes fusions of the leakage areas or CNV may occur. We think that this is a real biological phenomenon that arises from the strong effect of *Ifnar1* deficiency on immune cell activity and subsequent angiogenic responses. We have constantly and exclusively noticed these fused CNV in *Ifnar1* deficient animals. This is in full agreement with the hypothesis and findings of our study and therefore, we cannot exclude these important findings in the revised manuscript.

Response to reviewers Figure: Representative infrared fundus images of laser coagulation experiments for all different mouse strains and the treatment study presented in our manuscript. Relatively equal distributions without fusions can be noticed.

2. The inflammation was only analyzed at d3, even though the vascular changes are invariably only observed at d14. If the differences in IBA1+cells at d3 were responsible for the vascular phenotype, you would expect the vascular phenotype to be different at d7. It seems therefore likely that the IFNAR1-dependent differences in inflammatory phenotype that are responsible for the vascular changes occur at a later stage, or they should at least be analyzed additionally at a later time point.

Response:

We fully agree with the reviewer. To close this gap between early inflammation and late CNV changes we have now performed additional analyses in all three analysis groups (*Ifnar1^{-/-}*, *IFN-β* therapy and *Cx3cr1^{CreER}:Ifnar1^{flox/flox}* animals) with retinal sections at day 3 and RPE/choroidal flat

mounts at day 7. These data can be found in the Expanded View Figures EV1, EV2 and EV3. They show that mononuclear phagocytes in *Ifnar1*^{-/-} and *Cx3cr1*^{CreER}:*Ifnar1*^{fllox/fllox} animals are longer activated and accumulate in the subretinal space. These images also show a faster clearance of reactive microglia/macrophages in the laser spots of IFN-β treated mice.

3. The analysis of the *Cx3cr1*CreER:*Ifnar1*fl/fl mice does not allow to specifically analyze microglial IFNAR1 expression per se in this model, as the lesion was made in the choroid that contains numerous resident macrophages that likely express *cx3cr1* and also have a slow turnover.

Response:

We agree with the reviewer that potentially long-lived resident *Iba1*⁺ cells e.g. choroidal macrophages could also contribute to the effect seen when using the *Cx3cr1*^{CreER}:*Ifnar1*^{fllox/fllox} model. To overcome the potential confusion when using only the term “microglia”, we have now throughout the manuscript used the terms “*Iba1*⁺ cells”, “microglia/macrophages” or “mononuclear phagocytes”. We think that all three terms are scientifically correct and we have also included a reference that gives some information on non-microglial mononuclear cells in the eye: McMenamin PG (1999) Dendritic cells and macrophages in the uveal tract of the normal mouse eye. *Br J Ophthalmol* 83:598-604. However, this reference and also other papers we found did not really define how long these cells could potentially live or when they are repopulated.

4. Also the deletion of *Ifnar1* was not controlled for in the tissue and the different macrophage populations in the hands of the authors and a Cre-expressing control is missing.

Response:

We have now performed additional experiments to demonstrate *Ifnar* deletion in *Ifnar1*^{-/-} and *Cx3cr1*^{CreER}:*Ifnar1*^{fllox/fllox} animals (Appendix Supplementary Figure S3). Therefore, we have used genomic PCR to demonstrate the genomic deletion of *Ifnar1* exon 10 (Appendix Fig.S3A), Western blot analysis of total retinal extracts of *Ifnar1*^{-/-} and *Cx3cr1*^{CreER}:*Ifnar1*^{fllox/fllox} animals using a specific anti-*Ifnar* antibody (Appendix Fig.S3B), and immunohistochemical staining of sections from *Ifnar1*^{-/-} and *Cx3cr1*^{CreER}:*Ifnar1*^{fllox/fllox} mice using the same specific anti-*Ifnar* antibody together with *Iba1* (Appendix Fig.S3A). We also tried to perform *ex vivo* isolation of *Iba1*⁺ cells with MACS and thereafter perform FACS analysis but this experimental set up repeatedly failed because of limitations in total *Iba1*⁺ cell numbers and obviously incompatibility of the antibody for FACS.

5. IFNAR1 expression is not analyzed in the model and other cell types that might express IFNAR1 and mediate the therapeutic effect of IFN beta are therefore not.

Response:

This question was already partially covered in the response to question 4: Appendix Fig.S3B shows protein expression levels of *Ifnar* in total retinal extracts of wild-type mice, a lack of expression in *Ifnar1*^{-/-} mice and reduced levels in *Cx3cr1*^{CreER}:*Ifnar1*^{fllox/fllox} animals. Furthermore, we demonstrate a good co-localization of *Ifnar1* with *Iba1* in retinal sections of wild-type mice, which is lost when *Ifnar1*^{-/-} and *Cx3cr1*^{CreER}:*Ifnar1*^{fllox/fllox} animals were analyzed (see Appendix Supplementary Figure S3C). These data show that the therapeutic effect of IFN-β is very likely mediated by mononuclear phagocytes that express *Ifnar1*.

6. Abstract and discussion:

The authors state in several sentences to have analyzed the specific role of microglial IFNAR1. These conclusions cannot be drawn from the presented data as there are likely other resident macrophages with a slow turnover and a decent *Cx3cr1* expression that participate in the

inflammation associated with the laser-injury. (...interferon- β ; signaling in the retina accelerates microgliosis ... enhanced early microglia reactivity in lesion areas ... IFNAR in microglia only)

Response: Please see response to question 3

To overcome the potential confusion when using only the term “microglia”, we have now throughout the manuscript used the terms “Iba1+ cells”, “microglia/macrophages” or “mononuclear phagocytes”.

7. Introduction:

The previous study by Yasukawa et al on interferon and CNV in rabbits is not mentioned in the introduction. A previous study on laser-induced CNV in monkeys is not cited anywhere in the manuscript (Nippon Ganka Gakkai Zasshi. 1995 May;99(5):571-81. The effect of interferon-beta on experimental choroidal neovascularization. Tobe T1, Takahashi K, Ohkuma H, Uyama M.) There also is an interesting case report about an interferon beta and CNV that is not cited at all (Retina. 2006 Nov-Dec;26(9):1091-2. Resolution of activity (choroiditis and choroidal neovascularization) of chronic recurrent punctate inner choroidopathy after treatment with interferon B-1A. Cirino AC1, Mathura JR Jr, Jampol LM.)

Response:

The publication by Yasukawa et al. was already cited in the first submitted manuscript but was included in the discussion and not introduction. Since the Yasukawa article is well suited to discuss common and divergent findings related to our study we still think that discussion is the better place than introduction.

We have included the two other mentioned article in the introduction, the text reads as follows: “Previous studies have shown beneficial effects of IFN- β treatment on laser-induced CNV in rabbits (Kimoto et al, 2002) and monkeys (Tobe et al, 1995) by influencing the function of RPE and endothelial cells. Here, we demonstrate a pivotal effect of Ifnar/IFN- β -signaling in retinal microglia and macrophages cells that reduce the inflammatory and angiogenic events and thereby limit the development of CNV lesions.”

Both papers were also discussed in conjunction with the case report by Cirino et al.: “In accordance with this hypothesis, IFN- β treatment also ameliorated laser-induced CNV in rabbits (Kimoto et al, 2002) and monkeys (Tobe et al, 1995). Of note, a patient with multiple sclerosis and punctate inner choroidopathy could significantly profit from systemic IFN- β therapy and was subsequently free of active CNV (Cirino et al, 2006).

8. Results:

Figure 1: In the experiments represented in Figure 1 the authors analyzed IBA1+cell infiltration (d3) and CNV (d3, d7, d14) in wildtype and Ifnar-/- mice. The results presented in figure 1 do not allow a differentiation between microglia, infiltrating inflammatory macrophages or choroidal resident macrophages. The authors need to be more precise in their wording as the cells they count in the lesions likely are a mixture of all three cell types.

Response: Please see response to question 3

To overcome the potential confusion when using only the term “microglia”, we have now throughout the manuscript used the terms “Iba1+ cells”, “microglia/macrophages” or “mononuclear phagocytes”.

9. The authors counted amoeboid IBA1+ cells and crossing points. The total number of IBA1+ cells needs also to be shown.

Response:

The total number of Iba1+ cells was counted and is now included in Appendix Figure S1. It was not statistically different between the mouse groups analyzed.

10. Why were the IBA1+ cells only quantified at d3, when the vascular changes are only apparent at d14? A later analysis of IBA1 cell numbers and phenotype is necessary if the authors think that the Iba1 cell phenotype is the reason for the vascular differences.

Response: Please see response to question 2

11. The CNV lesions are very close to the optic nerve. In panel N and P, CNV bridging two to three laser lesions are visible. When laser lesions are two close to each other a CNV lesion often forms between two laser lesions, which is bigger than the addition of two separate lesions. I would therefore suggest not to include bridging CNV lesions, as their size is also influenced by the aleatory distance to the neighboring CNV.

Response: Please see response to question 1

12. Also I am not sure I understand the quantification method of angiography fluorescence by choosing two regions of interest (ROI) within and one ROI outside the laser spots. Are the CNV size quantifications per/Impact? If so, how were confluent CNVs quantified?

Response

The CNV quantification was done as mean values per eye. Two ROIs inside the leakage and one lesion in the background are illustrated in the next figure (for review purpose only) (A). To localize the laser spots, infrared images taken in the same positions were used as reference (B).

Response to reviewers Figure: Representative fundus fluorescein angiography and infrared fundus images of laser coagulation experiments delineating the ROI and quantification methods.

The text in the Materials and Methods section reads as follows:

“To quantify the laser-induced vascular leakage the pixel intensity was measured in two regions of interests (ROI) within and one ROI outside each laser spot using the image processing program ImageJ (NIH). Because three laser spots were induced per eye, we quantified the pixel intensity of six ROI within and three ROI outside the fluorescein leakages. After averaging the values and subtracting the background, one data point represented the mean laser-induced leakage per eye. Eyes were excluded from the analysis in case of choroidal hemorrhages and when laser lesions completely fused with each other or the optic nerve head.”

13. Why did the authors use lectin as a vascular stain, as it also marks activated IBA1 cells?

Response

Iba1-lectin co-staining of retinal cryosections and RPE/choroidal flat mounts revealed that lectin does not co-stain mononuclear phagocytes similarly as with Iba1 (see Expanded View Figures EV1, EV2, EV3). Staining with lectin is a commonly used and widely cited technique in the field to detect neovessels and there is no superiority of FITC-dextran staining in our opinion. Moreover, dextran-staining requires an elaborate animal perfusion to reach all blood vessels. This procedure is quite error-prone and not easy to standardize. For the reviewer, we have performed dextran/lectin double stainings of mouse RPE/choroidal flat mounts 7 days after laser coagulation. This image shows that lectin is well suited and that it seems superior to dextran as it stains more vessels and can be better quantified.

Response to reviewers Figure: Representative co-labeling of lectin and dextran in mouse RPE/choroidal flat mounts 7 days after laser coagulation.

14. Figure 2: In the experiments represented in Figure 2 the authors analyzed IBA1+cell infiltration (d3) and CNV (d3, d7, d14) in mice treated with IFN beta. Again the IBA1 cell number and phenotypes were only analyzed at d3, when the vascular differences only appeared 11days later.

Response: Please see response to question 2

15. Also the total number of IBA1+cells is again missing.

Response: Please see response to question 9

16. Panel N reveals confluent CNVs. Also the laser spots are sometimes distributed equally around the optic nerve (M) sometimes only to one side of the ON (N).

Response: Please see response to question 1

17. The authors also measured edema formation in this set of experiments. Edema is a fluid accumulation within the tissue (either intracellular or intercellular). How do the authors distinguish between edema and infiltrating cells (IBA1+ cells and proliferating endothelial cells) in the OCT images?

Response:

Since we do not have ultimate proof that the structure seen in OCT is edema/fluid and since these data are not fundamental for the story, we have excluded this figure subpanel from the revised paper.

18. Figure 3: In Figure 3 the authors analyzed IBA1+ cell infiltration (d3) and CNV (d3, d7, d14) in *Ifnar1*^{fl/fl} mice and tamoxifen-treated *Cx3cr1*^{CreER}:*Ifnar1*^{fl/fl} mice. The authors induced Cre expression 28 and 26 days prior to the laser lesion which should permanently delete the *Ifnar1* gene in cells that express high levels of *Cx3cr1*. 26 days after the TAM treatment cells with a high turnover, such as monocytes will again express *Ifnar*. The authors therefore state that the experimental mice only have a lack of *Ifnar* in microglial cells. This is however likely incorrect, as resident macrophages in choroid and ciliary body also have a very slow turnover. They likely participate in the laser-induced inflammation (they are actually closer to the burn than microglial cells) and will likely still lack (at least in part) *Ifnar* expression. It is therefore not possible to decipher the role of *ifnar* in only microglial cells using these mice.

Response: Please see response to question 3

19. The efficiency of *Ifnar* deletion in *Cx3cr1* expressing cells with high and low turnover were not analyzed in the hands of the authors and no data from *Cx3cr1* cre expressing mice is presented.

Response: Please see response to question 4

21. The laser spots are again very close to the optic nerve (one CNV in panel N seems to actually merge with the optic nerve). They are again not distributed equally, sometimes being only on one side of the optic nerve, sometimes all around.

Response: Please see response to question 1

Referee #2

Major comments

1. The authors should show cross sections of the CNV lesions induced laser coagulation as well as cross sections stained with Iba1.

Response:

This comment was very valid and also refers to question 2 of reviewer 1. To close this gap between early inflammation and late CNV changes we have now performed additional analyses in all three experimental groups (Ifnar1^{-/-}, IFN- β therapy and Cx3cr1^{CreER}:Ifnar1^{flox/flox} animals) with retinal cross sections at day 3 and RPE/choroidal flat mounts at day 7. These data can be found in the Expanded View Figures EV1, EV2 and EV3. They show that mononuclear phagocytes in Ifnar1^{-/-} and Cx3cr1^{CreER}:Ifnar1^{flox/flox} animals are longer activated and accumulate in the subretinal space compared to their respective controls. These images also show a faster clearance of reactive microglia/macrophages in the laser spots of IFN- β treated mice.

2. It would be informative to include the expression level of Ifnar in wt, Ifnar1^{-/-} and Cx3cr1CreER:Ifnar1f1/fl mice in the retina using immunoblot and immunohistochemistry

Response:

We fully agree with this comment that partially relates to question 4 of reviewer 1. We have now performed additional experiments to demonstrate Ifnar deletion in Ifnar1^{-/-} and Cx3cr1^{CreER}:Ifnar1^{flox/flox} animals (Appendix Supplementary Figure S3). Therefore, we have used genomic PCR to demonstrate the genomic deletion of Ifnar1 exon 10 (Appendix Fig.S3A), Western blot analysis of total retinal extracts of Ifnar1^{-/-} and Cx3cr1^{CreER}:Ifnar1^{flox/flox} animals using a specific anti-Ifnar antibody (Appendix Fig.S3B), and immunohistochemical staining of sections from Ifnar1^{-/-} and Cx3cr1^{CreER}:Ifnar1^{flox/flox} mice using the same specific anti-Ifnar antibody together with Iba1 (Appendix Fig.S3A). We also tried to perform ex vivo isolation of Iba1⁺ cells with MACS and thereafter perform FACS analysis but this experimental set up repeatedly failed because of limitations in total Iba1⁺ cell numbers and obviously incompatibility of the antibody for FACS.

3. Acute laser injury may not have relevance to a chronic degeneration disorder, including age related macular degeneration.

Response:

We are aware of the fact that the murine laser-coagulation model has some limitations especially related to the aspect of aging. However, our hypothesis for this work was that Ifnar signaling has a potential therapeutic effect by limiting retinal inflammation and thereby indirectly also choroidal neovascularization, both typical hallmarks of AMD. These aspects were well covered by the model and in addition we had the option to use several different genetically modified animals. Throughout the manuscript we were very careful not to over-interpret our findings.

Minor comments:

1. Laser-treated wild-type mice are labeled as C57BL6/J in Figure 1 and as control in Figure 2 - please make it consistent.

Response:

The labels in Figure 2 were changed to 'C57BL6/J'.

2. Figure 2C: the label "+IFN- β " is not necessary as it

Response:

The label "+IFN- β " in Figure 2C was removed.

Referee #3

1. The quantification of amoeboid shaped cells in Fig 1B/D will need to be elaborated upon in the methods section as it strikes me as a very subjective way of data analysis.

Response:

For quantification of mononuclear phagocyte morphology, we followed a method using a grid system to determine the number of grid crossing points published by Chen et al., 2012, *Glia* 60:833-42 (cited herein). The same method has been published by our group in a paper by Scholz et al., *J Neuroinflammation*. 2015 12:201. For quantification we used at least three different laser lesions of three independent animals. This information is added in the Materials and Methods section of the revised version as follows “The total number of Iba1+ cells and the number of amoeboid-shaped cells were counted within a circular region of 200 μ m diameter around the laser spots. Cellular morphology was analyzed using a grid system to determine the number of grid crossing points per cell (n=40-70 cells; from at least 3 different retinas per group) (Chen et al, 2012).”

2. What is the homology of human IFN-beta to mouse IFN-beta and are there differences in bioactivity between the two? Human IFN-beta only has 60% homology to mouse IFN-beta...would there be differences in therapeutic readout if the authors used mouse IFN-beta?

In this translational work, we made use of human IFN- β because of its relevance as a known human therapeutic compound. Human interferon- β has been widely used in previously published reports *in vitro* and in animal models, including *in vitro* assays with bovine RPE cells and *in vivo* experiments with rabbits (Yasukawa et al., *IOVS* 2002, 43:842; this paper is also cited in this manuscript). Nevertheless, we have performed additional *in vitro* experiments with microglial cells for the reviewer’s information (see Figure below). Real-time qPCR data on classical interferon- β target genes (myxovirus resistance 1 and 2, Mx1 and Mx2) showed that human interferon- β has good biological activity in the murine BV-2 cells line, albeit at lower levels than murine interferon- β . Interestingly, a human SV40 immortalized microglia cell line showed nearly the same induction of Mx1 after human interferon- β treatment than BV-2 cells did. Therefore, given the dose and repeated administration of human interferon- β in our model, we think that there would be no major differences in the therapeutic read out, which was already highly significant as can be seen from Figure 2.

Response to reviewers Figure: Representative real-time qRT-PCR data on classical interferon- β target genes (Mx1 and Mx2) after stimulation of murine BV-2 microglia and human SV40 immortalized microglia with human interferon- β .

3. Figure 3 should have included a control group of the Cx3cr1CreER mice on their own. It is widely accepted that Cre recombinase can have biological effects and toxicity when expressed in cells and this control would be important to include in the figure. This control would need to be used in all sub-sections of Figure 3. It will markedly strengthen the paper.

Response:

This comment was very valid and we have now included the analysis of Cx3cr1^{CreER} mice in all subpanels of Figure 3 as well as in Figure EV3. Moreover, the text was changed to include this important control at all relevant passages of the manuscript.

2nd Editorial Decision

30 March 2016

Thank you for the submission of your revised manuscript to EMBO Molecular Medicine. We have now received the enclosed reports from the referees that were asked to re-assess it. As you will see the reviewers are now supportive and I am pleased to inform you that we will be able to accept your manuscript pending final editorial amendments.

I look forward to receiving a new revised version of your manuscript as soon as possible, and within 2 weeks.

***** Reviewer's comments *****

Referee #1 (Comments on Novelty/Model System):

while the effect of IFN beta on CNV was previously known this paper shows its important role on mononuclear phagocytes using adequate genetic mouse models. I am not sure this results will directly translate into a new medical approach as the major pharmacological culprits of IFN stability and necessary slow release formulations seem to me to be a remaining major challenge.

Referee #1 (Remarks):

The additional experiments (analysis at intermediate time points, the verification of gene deletion and the analysis of Cre expressing controls) and the correct wording ("Iba1+ cells", "microglia/macrophages" or "mononuclear phagocytes") in the revised manuscript respond to all my concerns. I think this is a very nice study that highlights the importance of IFN signaling in resident mononuclear phagocytes. I would still advise the authors to place their laser impacts at greater distance from each other and the optic nerve in their futur studies (it is feasible in mice, we do it all the time).

Referee #2 (Comments on Novelty/Model System):

Due to lack of animal models that approximate macular degeneration, laser injury may be acceptable.

Referee #3 (Comments on Novelty/Model System):

This manuscript has been substantially improved and authors have addressed comments in detail.

Corresponding Author Name: Prof. Dr. Thomas Langmann

Manuscript Number: EMM-2015-05994